# The odorant metabolizing enzyme UGT2A1: Immunolocalization and impact of the modulation of its activity on the olfactory response

**Fabrice Neiers** ⓘ *, **David Jarriault, Franck Menetrier, Philippe Faure, Loïc Briand, Jean-Marie Heydel** ⓘ *

Centre des Sciences du Goût et de l'Alimentation, AgroSup Dijon, CNRS, INRAE, Université Bourgogne Franche-Comté, Dijon, France

* jean-marie.heydel@u-bourgogne.fr (JMH); fabrice.neiers@u-bourgogne.fr (FN)

## Abstract

Odorant metabolizing enzymes (OMEs) are expressed in the olfactory epithelium (OE) where they play a significant role in the peripheral olfactory process by catalyzing the fast biotransformation of odorants leading either to their elimination or to the synthesis of new odorant stimuli. The large family of OMEs gathers different classes which interact with a myriad of odorants alike and complementary to olfactory receptors. Thus, it is necessary to increase our knowledge on OMEs to better understand their function in the physiological process of olfaction. This study focused on a major olfactory UDP-glucuronosyltransferase (UGT): UGT2A1. Immunohistochemistry and immunogold electronic microscopy allowed to localize its expression in the apical part of the sustentacular cells and originally at the plasma membrane of the olfactory cilia of the olfactory sensory neurons, both locations in close vicinity with olfactory receptors. Moreover, using electroolfactogram, we showed that a treatment of the OE with beta-glucuronidase, an enzyme which counterbalance the UGTs activity, increased the response to eugenol which is a strong odorant UGT substrate. Altogether, the results supported the function of the olfactory UGTs in the vertebrate olfactory perireceptor process.

## Introduction

Xenobiotic metabolizing enzymes (XMEs) like cytochrome P450 (CYP), carboxylesterases (CE), glutathione transferases (GST), or UDP-glucuronosyltransferases (UGT) are involved in the detoxification of exogenous and endogenous active molecules by catalyzing their biotransformation in inactive metabolite easily eliminable from the body. Phase I enzymes (CYP, CE, . . .) catalyze the formation of a functionalize chemical group (-OH, -COOH, . . .) to the xenobiotic, Phase II (GST, UGT, etc) catalyze the subsequent conjugation of polar group (glutathione, glucuronic acid, etc). Mainly localized in the liver, their presence and activity toward volatile odorant substrates has been also evidenced at a high level in the olfactory tissues even

number ANR-18-CE92-0018-01 https://anr.fr FN: Agence Nationale de la Recherche ANR-16-CE21-0004-01 https://anr.fr The funders had no role in study design, data collection and analysis, decision to publish, or preparation of the manuscript.

**Competing interests:** The authors have declared that no competing interests exist.

higher for some isoforms [1–5]. There is an increasing body of proof demonstrating the main function of these odorant metabolizing enzymes (OMEs) in the physiology of olfaction. This role is also supported by the capacity of this large family of enzymes, particularly expressed in the olfactory epithelium (OE), to biotransform the same extraordinary large variety of odorants than those targeting the olfactory receptors. Enzymatic odorant metabolism catalyzed by OMEs has been involved (i) in odorants clearance from the perireceptor environment to terminate the signal and thus preserve the highest sensitivity of the detection and (ii) additionally, in the synthesis of metabolites generating additional stimuli potentially modulating or enriching the signal. This last function in signal modulation was initially investigated *in vivo* by observing the presence of odorant metabolites in mouse olfactory mucus after odorant exposure [6]. The authors also showed that the inhibition of the odorant metabolism impacted the pattern of glomeruli activated in the olfactory bulb and the ability of mice to discriminate odorants, suggesting a role of the metabolites in the signal. Moreover, we have recently demonstrated, using rat OE explant, with *ex vivo* headspace real-time mass spectrometry, the very fast synthesis (a hundred of milliseconds range) of metabolites showing odorant stimulus properties [7]. Using similar technic, *in vivo*, the presence of metabolites in the exhaled air was observed in human after odorant inhalation [8, 9].

The function of OMEs in signal termination has been also shown as critical in the olfactory peripheral process. We showed, on rat OE, that the olfactory response measured by electroolfactogram (EOG) increased when the phase I CYP activity responsible in the metabolism of the studied odorants was chemically inhibited [10]. Accordingly, we recently demonstrated, *in vivo*, that the disruption of the phase II GST dependent mammary pheromone metabolism [11, 12] led to increase its perception by newborn rabbits [13, 14]. We also evidenced the high interaction of human GST with a variety odorants [15].

UGT are major phase II detoxification enzymes, they catalyze the conjugation of UDP-glucuronic acid (glucuronconjugation) to diverse xenobiotics (pollutants, food additive, drugs, etc) or endobiotics (hormones, bilirubin, biliary acid, etc) consequently facilitating their elimination as hydrophilic metabolites. In the olfactory tissue, a high glucuronoconjugation activity toward diverse odorants has been demonstrated [5, 16, 17]. It has been shown using an olfactory cilia *in vitro* system that glucuronidated odorants metabolites did not elicit the production of cyclic AMP in the signal transduction pathway as did the parent odorants. More recently, using EOG in rat, we demonstrated that glucuronoconjugated metabolites of coumarin and quinoline triggered no olfactory response [10]. These studies support the involvement of UGTs in the olfactory signal termination. A pioneer study conducted in rat and bovine showed the expression and activity toward odorants of a particular UGT isoform named UGT2A1 [17]. UGT2 family includes the UGT2A and UGT2B families. Phylogenetic analysis shows that the UGT members are closely clustered between mammal species supporting an ancient diversification [18]. Rat UGT2A1 shares the highest sequence homology with human or murine UGT2A1 orthologs than any other UGTs including rat UGT2A2 the closest paralog. These observations support potential similar function between species for a same UGT including UGT2A1 between rat and human. This isoform is preferentially and highly expressed in the olfactory epithelium [16, 17, 19] and seems the most active olfactory UGT toward odorants [5]. UGT2A1 expression was initially localized in bovine OE mainly in the Bowman's gland (mucus gland) and at the apical region of the sustentacular cells [17]. In addition, mRNA UGT2A1 expression was evidenced in the Bowman glands, the sustentacular cells and olfactory sensory neurons using *in situ* hybridization [19]. The cellular localization of OMEs is of importance to support their role in the first step of the olfactory process. Indeed, since these enzymes modulate the availability of odorants or their metabolites for olfactory receptors, they are supposed to be active in the vicinity of the olfactory receptors carried by the olfactory sensory neurons [1].

In the present work, we used immunohistochemistry to determine UGT2A1 pattern in rat OE and electron microscopy to precisely localize the protein. Moreover, we investigated the impact of the inhibition of the UGT2A1 activity on the olfactory response toward eugenol, a highly glucuronidated odorant. Prior to EOG recording, the OE was treated *ex vivo*, by the beta-glucuronidase, an enzyme which catalyze the hydrolysis of glucuronoconjugated molecules, therefore expectedly annihilating UGT activity by releasing eugenol from the conjugate.

## Materials and methods

### Animals

Male Wistar rats (n = 24) used were 7 weeks old. They were housed in standard cages (900cm$^2$) in a 12L:12D lighting schedule (lights on at 6:30) with *ad libitum* access to water and food. The local, institutional and national guidelines and regulation regarding the applied methods, the care and experimental uses of the animals were followed. Rats were decapitated using guillotine without anesthesia to avoid any effect on the enzymes activity and their expression. All the precaution to alleviate suffering were taken (protocol validated by the Comité d'Ethique de l'Expérimentation Animale Grand Campus Dijon N˚105). All experimental protocols were conducted in accordance with ethical rules enforced by French law, and were approved by the local Ethical Committee of the University of Burgundy (Comité d'Ethique de l'Expérimentation Animale Grand Campus Dijon N˚105; C2EA grand campus Dijon, and by the French Ministère de l'Education Nationale, de l'Enseignement Supérieur et de la Recherche.

### Immunohistochemistry experiments

Two rats heads were fixed with formaldehyde solution 4% buffered pH 6.9 (1.00496, Merck, Darmstadt, Germany) for 48 h at room temperature. After decalcification with 10% Ethylenediaminetetraacetic acid disodium salt (Titriplex III, 1.08418, Merck, Darmstadt, Germany) in phosphate-buffered saline (PBS, P4417, Merck, Darmstadt, Germany) pH 7.4 for two weeks with daily changes of this solution, the specimens were dehydrated through a series of alcohols and toluene baths, then embedded in paraffin. Frontal five-micrometer thick sections were deparaffinized, rehydrated and stained immunohistochemically. An antigen pre-treatment step was carried out using high-temperature antigen unmasking techniques with target retrieval in citrate buffer pH 6.0 (S2369, Agilent, Santa Clara, USA) for 40min. Endogenous peroxidases were treated with blocking reagent (S2003, Agilent, Santa Clara, USA) for 10min at room temperature prior to equilibration in 0.05M Tris-HCl, 0.15M NaCl, 0.05% Tween 20, pH 7.6. Tissue sections were satured for 45 min with 10% donkey serum (D9663, Merck, Darmstadt, Germany) in antibody diluent (S0809, Agilent, Santa Clara, USA) to reduce non-specific binding. Incubation in the primary antibody was performed for overnight at 4˚C (1:50 from Santa Cruz Biotechnology, sc-244569, Dallas, USA). Negative controls were prepared by replacing the primary antibody with antibody diluent alone. Tissue sections were subsequently incubated for 45min at room temperature in a 1:200 dilution of the secondary antibody in diluent (donkey-anti-goat HRP from Santa Cruz Biotechnology, sc-2020, Dallas, USA). Immunohistochemical staining was performed using liquid DAB (diaminobenzidine) + Substrate chromogen system (K3468, Agilent, santa Clara, USA). Sections were conterstained with Mayer's hemalum solution (1.09249, Merck, Darmstadt, Germany). The slides were examined with a microscope Eclipse E600 equipped with plan fluor objectives. Images were acquired with a DS-Ri2 digital camera using the software NIS-Elements Basic Research (all from Nikon, Tokyo, Japan).

## Immunogold electron microscopy

Following anesthesia, two rats were perfused intracardially with fixative solution 4% paraformaldehyde (PFA) in Sorensen's buffer 0.1 M pH 7.3. Nasal tissues were dissected and the olfactory epithelium (OE) was isolated. OE was fragmented into small pieces post-fixed 2 h at 4˚C with 4% PFA and 0.1% glutaraldehyde. Tissues samples were then rinsed several times in Sorensen's buffer and were treated with 50 mM NH4Cl for 15 min at 4˚C to block aldehyde sites. The samples were quickly washed with water and then dehydrated in graded ethanol solutions and infiltrated with LR White resin at -20˚C with a progressive increase in the ratio of resin to ethanol. Polymerization was carried out with ultraviolet light for 15 h at -20˚C and terminated with daylight at room temperature for 48 h. Ultrathin sections (80 nm) were cut on a Reichert Ultracut E ultramicrotome (Leica Microsystems SAS, Nanterre, France) and were collected onto carbon-formvar-coated nickel grids. Grids (n = 25) were treated with Tris buffered saline (TBS) containing 0.05 M Tris-HCl pH 7.3, 150 mM NaCl supplemented with 0.1% glycine for 15 min to inactivate free aldehyde groups. They were then treated with in TBS containing 0.1% acetylated Bovine Serum Albumine (BSA-c, Aurion, Wageningen, Netherlands) and 10% normal donkey serum (Aurion, Wageningen, Netherlands). After rinsing with TBS+0.1% BSA-c, sections were incubated with the goat polyclonal antibody anti UGT2A1 (Santa Cruz Biotechnology, sc-244569, Dallas, USA) at the dilution of 1/50 in TBS+0.1% BSA-c for 90 min at room temperature. Negative controls were prepared by replacing the primary antibody with the dilution buffer only. Antigen-antibody reaction was detected with 10 nm gold-labelled donkey anti-goat IgG (Aurion, Wageningen, Netherlands) diluted 1/25 in TBS+0.1% BSA-c for 1 h at room temperature. The reaction was post-fixed using TBS+2% glutaraldehyde 5 min at room temperature and grids were contrasted with a solution of 3% uranyl acetate 10 min at room temperature. Then grids were examined under a Hitachi H7500 transmission electron microscope (Hitachi Scientific Instruments Co., Tokyo, Japan) operating at 80 kV and equipped with an AMT camera driven by ATM software (all from ATM, Danvers, USA).

## Modulation of the olfactory response experiments

**Stimuli.** Two odorants were chosen for their different chemical structures and their susceptibility to glucuronidation. Amyl acetate (AA) is not considered as a UGTs substrate whereas eugenol (EUG) is highly glucuronoconjugated. These compounds were diluted into distilled water while solubilized by DMSO. 10 μL of the diluted odorant solutions were deposited on filter papers contained in Pasteur pipettes. Odorant concentrations applied on olfactory mucosa were adjusted to ensure similar response amplitudes for both odorants ($10^{-3}$ M for amyl acetate and $10^{-2}$ M for eugenol).

**Beta-glucuronidase preparation.** Beta-glucuronidase from *Helix pomatia*, type H-1 (Sigma-Aldrich) was dissolved in mucosal saline solution (MS) just before the experiment: 10mg/mL; 45 mM KCl, 20 mM $KC_2H_3O_2$, 55 mM $NaCH_3SO_4$, 1 mM $MgSO_4$, 5 mM $CaCl_2$, 10 mM HEPES, 11 mM glucose, 50 mM mannitol, pH 7.4, 350 mOsm adjusted with mannitol. The composition of this solution has been described in previous studies (Negroni *et al.*, 2012) and was adjusted to match the composition of mucus as closely as possible.

**Electroolfactogram recordings (EOG).** Twenty rats were used for EOG recordings. Olfactory epithelium was exposed after decapitating rats and cutting their head on a sagittal plane. Both hemi-heads were used to probe olfactory responses on endoturbinates IIb and III either with vehicle or with β -glucuronidase treatment. Recordings were made with glass micropipettes of 6–8 μm diameter filled with a mucosal saline solution Reference electrode (Ag/AgCl) was placed in the olfactory bulb. Signal was amplified by an Axoclamp 200B amplifier (Axon Instruments, Molecular Devices, USA) and monitored with Axoscope (Axon

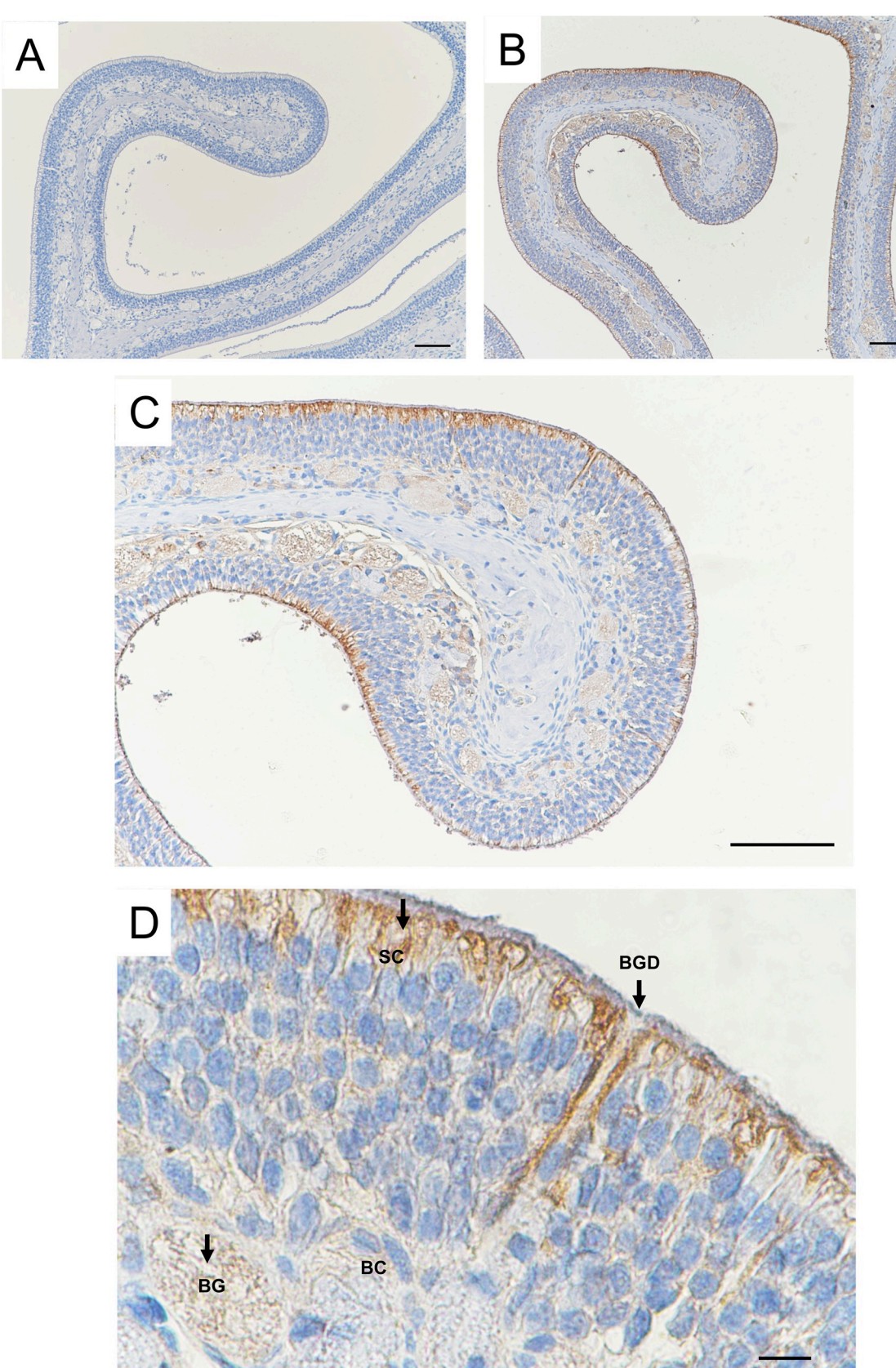

**Fig 1. Immunolocalization of UGT2A1 in the rat OE.** Distribution of UGT2A1 immunoreactivity using an anti-UGT2A1 antibody. (A) Control section in which the primary antibody was omitted. (B)(C) UGT2A1 staining is observable throughout the epithelium, mainly at the apical portion of the OE. (D) Higher magnification showing different cell types and structures, including the sustentacular cells (SC), Bowman gland (BG) and basal cells (BC). Staining was observed in the BG, Bowman gland duct (BGD) and in the apical portion of the epithelium including SC. The scale bar is 100 μm for (A), (B), (C) and 10 μm for (D).

Instruments, Molecular Devices, USA). Odorant air puff stimulation was delivered using a pressure controller for 200 ms at 200 mL/min inside a constant humidified airflow (1000 mL/min) blowing on the olfactory mucosa through a 7 mm-diameter tube. Odorant free stimulations were always tested prior to odorized ones. The OM was stimulated with each odorant before treatment with β-glucuronidase. Droplets of approximately 1 μL of either beta-glucuronidase or saline solution were delivered by capillarity onto the recorded area of endoturbinates IIb or III using glass micropipettes (∼5 μm in diameter). Odorant stimulations started one minute after application. Electrophysiological signal analysis was performed using customed Matlab routines. Only complete sequences of stimulations showing electrophysiological responses were conserved for subsequent analysis. Signal amplitude measured for odorant free stimulations was subtracted from odorant elicited signals. Depolarization amplitude, depolarization speed, fast (from 90 to 50%) and slow repolarization (from 50 to 10%) speeds of EOG responses were measured. A ratio was calculated between values obtained before and after treatment with MS or β-glucuronidase solution (fold change after treatment EOG amplitude / before treatment EOG amplitude) following a procedure published elsewhere [20–22]. Ratio values over two standard deviations around the mean were excluded (2 out of 38 in vehicle group and 2 out of 40 in β-glucuronidase group). A 2-way ANOVA (Prism, GraphPad Software, USA) was used to determine statistical differences between odorants and characterise the treatment effect. Post hoc multiple comparisons were performed using Bonferroni tests.

## Results

### UGT2A1 is expressed in the close vicinity of olfactory receptors

In immunohistochemistry experiments, UGT2A1 expression was observed throughout the OE in comparison with the control without staining (Fig 1A and 1B). The highest magnification (Fig 1C and 1D) confirmed the localization of UGT2A1 in the supranuclear portion of the sustentacular cells impressing a marked staining all along the apical portion of the OE. In addition, a slight staining was observed in the bowman gland and duct. Immunogold electron microscopy was used to focus on the localization of UGT2A1 at the apical portion of the OE. Using immunogoldstaining, UGT2A1 was clearly localized in the olfactory knob of the sensory neurons (Fig 2B, 2C and 2D), precisely to the plasma membrane of the olfactory cilia (Fig 2E and 2F). In addition, UGT2A1 staining was observed in the endoplasmic reticulum of the sustentacular cells (S1 Fig).

### Disruption of glucuronidation affects the olfactory response

The impact of glucuronidation on odorant elicited electrical activity of the olfactory mucosa was tested using EOG recordings on *ex vivo* preparations from rats. Beta-glucuronidase was applied to the OM to test the effect of "deglucuronidation" on the response to two odorants eugenol and amyl acetate, respectively with high and no susceptibility to glucuronidation. We compared the olfactory responses to the odorants after treatment with MS containing beta-glucuronidase and MS alone (Fig 3). Local treatment of the OM with the MS slightly decreased the response amplitude to both odorants, mainly due to a dilution effect of the mucus (respectively -20.3 ± 0.17% and -15.4 ± 0.12% for amyl acetate and eugenol; 2-ways ANOVA post hoc

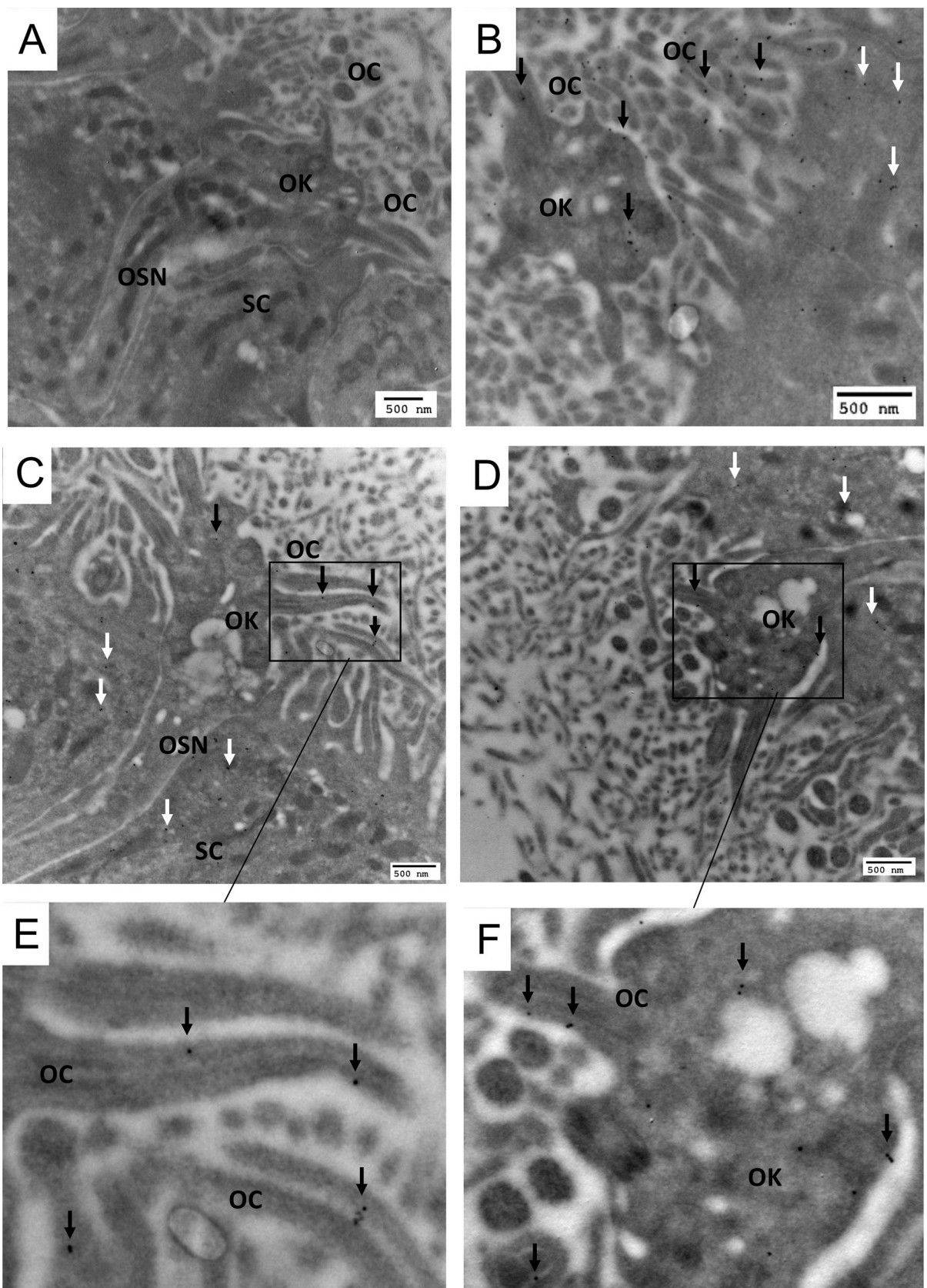

**Fig 2. Electron microscopy immunogold localization of UGT2A1 in the rat OE.** (A) control section in which the primary antibody was omitted and showing different cell types and structures, including the olfactory sensory neurons (OSN), sustentacular cell (SC), olfactory knob (OK) and olfactory cilia (OC). (B)(C)(D)(E)(F) UGT2A1 was localized in the OK particularly to the OC plasma membrane (black arrows). (B)(C)(D)UGT2A1 was also observed in the supranuclear portion of the SC (white arrows).

comparisons: $p < 0.01$). When beta-glucuronidase was added to MS no significant change was measured on the amplitude of responses to amyl acetate which is not glucurono-conjugated ($-30.7 \pm 0.16\%$). Conversely an increase of the amplitude of responses to eugenol which is glucurono-conjugated was observed when beta-glucuronidase was added to MS ($+13 \pm 0.14\%$) (Fig 3A and 3B; 2-ways ANOVA post hoc comparisons: respectively $p = 0.498$ and $p = 0.037$). EOG signal kinetics were not affected by beta-glucuronidase treatment indicating that enzymatic activities in the olfactory transduction cascade were not affected by the treatment.

## Discussion

With regard to the recent highlight on the function of OMEs in vertebrate olfaction, a deeper characterization of the odorant metabolic capacity of the olfactory tissues will help to decipher the complex equilibrium occurring in the peripheral olfactory process.

Alike other XMEs, OMEs are endoplasmic reticulum membrane bound (UGT, GST, CYP) or cytosolic enzymes (GST, sulfotransferase, carboxylesterase, alcohol dehydrogenase, etc.). Here, our results, for the first time, strongly suggest the original localization of UGT2A1 to the plasma membrane of the OE olfactory cilia. The localization of membrane bound OMEs in the OE plasma membrane cells would support the fast and efficient metabolic rate toward odorants allowing a direct contact with odorants contain in olfactory mucus. Such unconventional addressing for these enzymes has been observed for CYP and GST [23–27] but never investigated in olfactory tissues. Interestingly, a recent work demonstrated for the first time the plasma membrane localization and activity of a UGT in mammalian cells [28]. The authors investigated the expression and activity of UGT1A8 in the HT29-MTX cells which are human intestinal cellular model. They observed the presence of UGT1A8 in the basal and lateral parts of the plasma membrane and demonstrated a high glucuronidation rate toward substrates applied outside the cells. Topologically, the active site of the enzyme may face the intracellular space as observed conventionally in the endoplasmic reticulum [29] since it has been shown that the external addition of the co-substrate uridine-5'-diphosphoglucuronic acid did not results in an increase of the conjugate synthesis either in the supernatant or the cytosol [28]. On the whole, these results support a functional role of UGT2A1 in the cilia plasma membrane, a localization which would favor an efficient and faster glucuronoconjugation of odorants present in the mucus.

Such localization comes in addition to the presence and activity of OMEs in the olfactory mucus [6, 14], in direct contact with odorants, also supporting the functional role of OMEs in the modulation of the olfactory signal. However, only soluble enzymes with free access to their co-substrates can be active in the mucus. It is unlikely that UGT could be present in the mucus because its activity has been shown to be dependent to the surrounding phospholipids in the membrane [30–32]. Accordingly, proteomic studies did not evidence UGT protein in the mucus in rat or human [4, 33–35], while, as confirmed by our results, UGT2A1 has been detected in olfactory cilia proteome [36, 37]. One should note that cross-reactivity of the antibodies with the UGT2A2 variant cannot be ruled out. UGT2A2 was also identified in the olfactory cilia proteome [36, 37]; It was shown much less expressed and active than UGT2A1 [38].

Eugenol is a well-known aromatic phenolic compound which is a very good substrate of UGT2A1 [5, 16]. Inhibition of OMEs activity has been a useful tool to demonstrate their

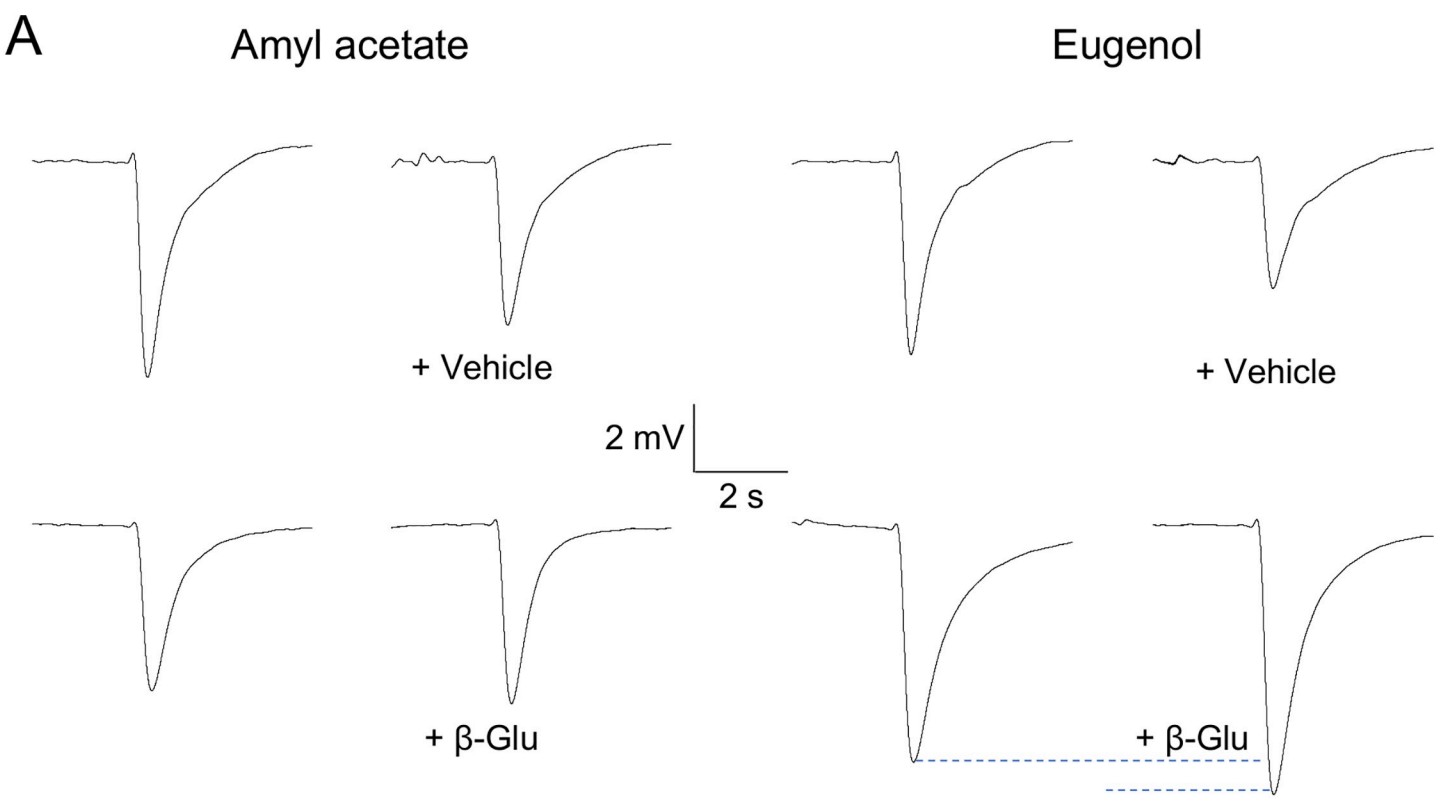

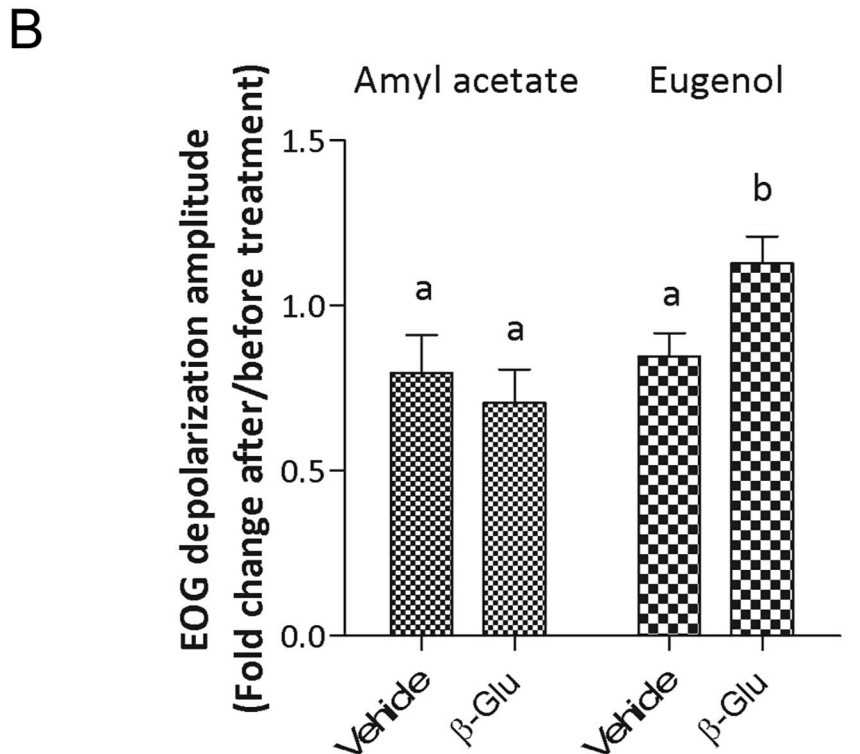

**Fig 3. Beta-glucuronidase treatment enhances OM response to eugenol but not to amyl acetate.** (A) Typical EOG responses to amyl acetate and eugenol before and after treatment with mucosal solution (Vehicle) alone or containing β-glucuronidase (10 mg/mL). (B) Effect of β-glucuronidase treatment on amplitude of EOG signals recorded from rat olfactory mucosa (Vehicle: n = 18 olfactory mucosa; β-glucuronidase: n = 19 olfactory mucosa, from 19 rats). Mean ± SEM values are expressed as fold change from baseline responses (prior to treatment). Bars with the same letters are not significantly different (2-way ANOVA with Bonferroni post-hoc comparisons, $P < 0.01$).

function in the olfactory response or perception toward specific odorants [6, 10, 13, 14]. Here, we proposed a new and original approach consisting to modulate the activity of UGT2A1 toward eugenol by adding the glucuronidase enzyme on the surface of the OE. The glucuronidase solubilized in the olfactory mucus was expected to exert its activity in this medium, a diffusion or transport in the intracellular space is unlikely. According to the localization of UGT2A1 in the membrane of the olfactory cilia of the olfactory sensory neurons, we hypothesized that, after exposure to eugenol, the action of the glucuronidase in the mucus would catalyze the hydrolysis of the eugenol-glucuronide resulting in the release of eugenol. Indeed, in these experimental conditions, using EOG, we observed that the olfactory response toward eugenol increased in presence of glucuronidase, while no effect of the treatment was observed on isoamyl acetate response, a non glucuronoconjugated odorant. We observed that the counterbalance of UGTs activity by glucuronidase activity resulted in an increase of the odorant EOG response supporting the function of UGTs in the termination of the signal. These results also confirmed that glucuronoconjugated eugenol was present in the mucus since the glucuronidase impacted eugenol EOG response. Accordingly, as observed in the histological experiments, UGT2A1 was also present and probably active also within the sustentacular cells and Bowman glands, therefore our results may suggest that after intracellular metabolism, phase II glucuronide metabolites were excreted in the olfactory mucus for their elimination. Thus, the release of eugenol evidenced by EOG after glucuronidase treatment may be a result of extracellular (cilia plasma membrane UGT2A1) and intracellular eugenol metabolism. Interestingly, beta-glucuronidase activity has been observed in the rodent olfactory epithelium [39]. The presence of beta-glucuronidase suggests a feedback process on the glucuronidation of xenobiotics entering the nasal cavity including odorants.

In the detoxification metabolic process, UGTs are major enzymes, here within the olfactory process, by disturbing their function in odorant clearance, we were able to observe a consecutive increase in the EOG response. Interestingly, with regard to the importance of UGT2A1 in the protection of the respiratory and aerodigestive track, UGT2A1 variants as splicing variants have been identified playing a potential role in UGT2A1 activity [40, 41]. Moreover, UGT2A1 expression is regulated by microRNA [42]. Single-nucleotide polymorphism including non-synonymous substitutions [43] may potentially lead to different UGT2A1 catalytic activity toward odorants in human population. Since numerous odorants are directly or secondarily glucuronoconjugated, a better characterization of the olfactory glucuronoconjugation capacity including the study of UGTs polymorphism or regulation may help for a deeper understanding of the physiological or pathophysiological perireceptor process of olfaction.

## Supporting information

**S1 Checklist. The ARRIVE guidelines 2.0: Author checklist.**
(PDF)

**S1 Fig. Electron microscopy immunogold localization of UGT2A1 in the rat OE.** (A) control section in which the primary antibody was omitted and showing different cell types and structures, including the olfactory sensory neurons (OSN), sustentacular cell (SC), olfactory knob (OK) and olfactory cilia (OC). (B)(C)(E)(F) UGT2A1 was localized in the OK

particularly to the OC plasma membrane (black arrows). (D)(G)(H) UGT2A1 was also observed in the supranuclear portion of the SC and in the reticulum endoplasmic (white arrows).
(TIF)

**S1 Dataset.**
(PDF)

## Acknowledgments

The authors thank Jeannine Lherminier head of the INRA microscopy center for her technical assistance.

## Author Contributions

**Conceptualization:** Fabrice Neiers, Philippe Faure, Loïc Briand, Jean-Marie Heydel.

**Data curation:** Jean-Marie Heydel.

**Formal analysis:** Fabrice Neiers, David Jarriault, Philippe Faure, Jean-Marie Heydel.

**Funding acquisition:** Fabrice Neiers, Loïc Briand, Jean-Marie Heydel.

**Investigation:** David Jarriault, Franck Menetrier, Philippe Faure.

**Methodology:** Fabrice Neiers, Franck Menetrier, Philippe Faure, Jean-Marie Heydel.

**Project administration:** Jean-Marie Heydel.

**Resources:** Jean-Marie Heydel.

**Software:** Jean-Marie Heydel.

**Supervision:** Jean-Marie Heydel.

**Validation:** Fabrice Neiers, Jean-Marie Heydel.

**Visualization:** Fabrice Neiers, Jean-Marie Heydel.

**Writing – original draft:** Fabrice Neiers, Jean-Marie Heydel.

**Writing – review & editing:** Fabrice Neiers, David Jarriault, Philippe Faure, Loïc Briand, Jean-Marie Heydel.

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
