## [Decision Letter · Decision Letter 0]

8 Dec 2020

PONE-D-20-32295

The odorant metabolizing enzyme UGT2A1: Immunolocalization and impact of the modulation of its activity on the olfactory response.

PLOS ONE

Dear Dr. Heydel,

Thank you for submitting your manuscript to PLOS ONE. After careful consideration, we feel that it has merit but does not fully meet PLOS ONE’s publication criteria as it currently stands. Therefore, we invite you to submit a revised version of the manuscript that addresses the points raised during the review process.

We look forward to receiving your revised manuscript.

Kind regards,

Peng He, Ph.D

Academic Editor

PLOS ONE

Journal Requirements:

2. As part of your revision, please complete and submit a copy of the Full ARRIVE 2.0 Guidelines checklist, a document that aims to improve experimental reporting and reproducibility of animal studies for purposes of post-publication data analysis and reproducibility: https://arriveguidelines.org/sites/arrive/files/Author%20Checklist%20-%20Full.pdf (PDF). Please include your completed checklist as a Supporting Information file. Note that if your paper is accepted for publication, this checklist will be published as part of your article.

3. To comply with PLOS ONE submissions requirements, in your Methods section, please provide additional information on the animal research and ensure you have included details on (1) methods of sacrifice, (2) methods of anesthesia and/or analgesia, and (3) efforts to alleviate suffering.

Reviewers' comments:

Reviewer's Responses to Questions

**Comments to the Author**

1. Is the manuscript technically sound, and do the data support the conclusions?

Reviewer #1: Yes

Reviewer #2: Yes

Reviewer #3: Partly

2. Has the statistical analysis been performed appropriately and rigorously? 

Reviewer #1: Yes

Reviewer #2: Yes

Reviewer #3: Yes

3. Have the authors made all data underlying the findings in their manuscript fully available?

Reviewer #1: Yes

Reviewer #2: Yes

Reviewer #3: No

4. Is the manuscript presented in an intelligible fashion and written in standard English?

Reviewer #1: Yes

Reviewer #2: Yes

Reviewer #3: Yes

5. Review Comments to the Author

Reviewer #1: The research by Neiers et al. described the location of UGT2A1 in the apical part of the sustentacular cells and originally at the plasma membrane of the olfactory cilia of the olfactory sensory neurons, and showed that a treatment of the OE with beta-glucuronidase, an enzyme which counterbalance the UGTs activity, increased the response to eugenol which is a strong odorant UGT substrate. There are several issues that must be addressed before it can be accepted for publication.

1.What is the important theoretical and practical value of study UGT2A1 in understanding olfactory communication mechanism in rat? The authors should add this part to the discussion.

2.What are the evolutionary characteristics of UGT2A1 and is it related to other mammals? I suggest that the authors add relevant analysis, such as: constructing UGT phylogenetic tree.

3.The figures are not clear, I suggest the authors to improve the clarity.

Reviewer #2: Neiers et al report a study aiming at assessing the localization as well as the activity on olfaction of the odorant metabolizing enzyme UGT2A1. Peri-receptors events are a very exciting field in olfaction because it has only recently been admitted that odorant receptors are not the sole players in the perception of smell.

The papers is in many ways well written and starts with a thorough introduction which clearly puts the research into its context. Statistics are accurately performed and described. The main result is that UGT2A1 is localized at the plasma membrane of olfactory cilia. This finding is consistent with a fast and accurate modification of some odorants during our perception of smells.

To assess this effect, the authors modulated the activity of UGT2A1 toward an odorant which is sensitive to the glucuronidase enzyme. EOG recordings confirmed an increase of the response. Such an effect has not been observed on a non-sensitive odorant. Additional odorant controls could have been considered, while it does not appear mandatory.

The authors claim that the effect they observed is associated with a modulation of the olfactory response. I suggest downplaying this conclusion, as the real ‘olfactory response’ is much more complicated than a simple EOG recording. It implies various other aspects such as kinetics of binding, scavenging effect by odorant binding proteins or top-down regulation of the olfactory signal, etc. All in all, this is a pretty interesting paper which contributes to gain knowledge on the role of enzymes in the mechanism of olfaction.

Reviewer #3: In their manuscript "The enzyme UGT2A1, which metabolizes odors: immunolocalization and impact of the modulation of its activity on the olfactory response". Fabrice Neiers and his co-authors provide an anatomical and functional characterization of the rat UGT2A1 enzyme in the olfactory epithelium (OE).

For this purpose, the authors describe a study of the localization of UGT2A1 in the OE using IHC and electron microscopy, as well as an original inhibition approach based on the use of betaglucuronidase. Overall, the manuscript is clearly written, but some of the experiments seem to be preliminary, and the manuscript would be improved by a revision

The immunolocalization of UGT2A1 in rat OE is of great interest and the potential localization of UGT2A1 in the plasma membrane of olfactory cilia is very original and constitutes a potential breakthrough in the understanding of OE expression patterns and putative function. Unfortunately, the results obtained in the present study are not convincing enough to confirm such crucial finding. First, IHC experiments are highly reminiscent of the picture obtained by Lazard et al., 1991 where a similar labelling was described as “mainly in in subepithelial Bowman’s gland” and “in a superficial epithelial layer, probably corresponding to the apical cytoplasm regions of the secretory supporting cells”. The authors should therefore provide more detailed images with higher magnification of the apical region of the OE. More importantly, double IHCs with neuron-specific Ab are required to conclude on putative neuronal expression.

The same applies to the immunogold electron microscopy study: the authors claim the use of "stringent conditions" that remains to be explained, and that the number of images is too low, especially with regard to ciliary labelling, which must be combined with a control image in a much larger microscopic study (and please indicate the number of samples treated). Moreover the intensity of the IHC labeling is not comparable with the EM regarding the relative intensities (despite stringent conditions). Thus, while this part of the study is promising, the data presented in this section require further testing to confirm the authors' conclusions. ; the authors should also consider the fact that UGT2A2 could be detected by their analysis, as the Ab is unlikely to detect a specific isoform.

The electrophysiological approach suffers from the same caveat: the original idea of using betaglucuronidase coupled with EOG is brilliant, but this experiment requires much more analysis to conclude on the role of UGT2A1 in olfactory modulation. First traces in figure 3 are supposed to describe “typical EOG responses” but are not in line with the analysis depicted in the following graph (the number of tested animals is also not mentioned).The authors should consider traces averaging for an immediate and clear comparison of the different conditions. Moreover, in such a new a broader study should be conducted to validate the proposed approach: more odorants (guaiacol, phenylethanol, ethyl hexanol, or other aromatics…) and, more importantly, a large concentration range of betaglucuronidase to ensure its dose dependence effect on EOG responses.

Finally, in the discussion, the authors speculate on the potential role of UGT2A1 in modulating the olfactory response with, as I understand it, the enzyme facing the outer neuronal membrane that could buffers the amount of odorants for the OR to respond While interesting, this hypothesis is highly speculative because many of the cofactors necessary for the enzyme to function are absent from the OM. Additional experiments, such as in vitro analysis, could respond to this potential new enzymatic topology.

6. PLOS authors have the option to publish the peer review history of their article (what does this mean?). If published, this will include your full peer review and any attached files.

Reviewer #1: No

Reviewer #2: No

Reviewer #3: No

---

## [Author Response · Author response to Decision Letter 0]

25 Feb 2021

Dear Editor,

We strongly appreciate the interest of both PlosOne editor and the reviewers in our work. We sincerely thank the reviewers for the time they spent on our manuscript and for their interesting and constructive comments which helped us to improve the quality of our manuscript. In this paper, we propose a very focused histological and pharmacological approach supporting the function of the odorant metabolizing enzymes and UDP-glucuronosyltransferase in the modulation of odorant availability and EOG response.

We carefully analyzed the reviewer remarks and answered to them in the following text. 

Editor:

As requested:

- We added information in the “animals” section about the (1) methods of sacrifice, (2) methods of anesthesia and/or analgesia, and (3) efforts to alleviate suffering.

- We deleted our mention “data not shown” which was only a confirmation of previous results.

Reviewer #1: The research by Neiers et al. described the location of UGT2A1 in the apical part of the sustentacular cells and originally at the plasma membrane of the olfactory cilia of the olfactory sensory neurons, and showed that a treatment of the OE with beta-glucuronidase, an enzyme which counterbalance the UGTs activity, increased the response to eugenol which is a strong odorant UGT substrate. There are several issues that must be addressed before it can be accepted for publication.

1.What is the important theoretical and practical value of study UGT2A1 in understanding olfactory communication mechanism in rat? The authors should add this part to the discussion.

Authors answer: Thanks to the pertinent remark of the reviewer 1, we have realized that the rationale behind the focus on UGT2A1 was not clearly presented. We followed your advice to better present the reasons of the choice of UGT2A1 in the introduction. In addition, we insist on UGT2A1 interest in the conclusion. 

2.What are the evolutionary characteristics of UGT2A1 and is it related to other mammals? I suggest that the authors add relevant analysis, such as: constructing UGT phylogenetic tree.

Authors answer: We added a sentence, in the part related to UGT2A1 in the introduction, highlighting on the evolutionary characteristics of UGT2A1 based on phylogenetic trees previously published.

3.The figures are not clear, I suggest the authors to improve the clarity.

Authors answer: The quality of the figures was checked by PlosOne, you may need to open the blue link on the figure to access to the original one.

Reviewer #2: Neiers et al report a study aiming at assessing the localization as well as the activity on olfaction of the odorant metabolizing enzyme UGT2A1. Peri-receptors events are a very exciting field in olfaction because it has only recently been admitted that odorant receptors are not the sole players in the perception of smell. The papers is in many ways well written and starts with a thorough introduction which clearly puts the research into its context. Statistics are accurately performed and described. The main result is that UGT2A1 is localized at the plasma membrane of olfactory cilia. This finding is consistent with a fast and accurate modification of some odorants during our perception of smells. To assess this effect, the authors modulated the activity of UGT2A1 toward an odorant which is sensitive to the glucuronidase enzyme. EOG recordings confirmed an increase of the response. Such an effect has not been observed on a non-sensitive odorant. Additional odorant controls could have been considered, while it does not appear mandatory. The authors claim that the effect they observed is associated with a modulation of the olfactory response. I suggest downplaying this conclusion, as the real ‘olfactory response’ is much more complicated than a simple EOG recording. It implies various other aspects such as kinetics of binding, scavenging effect by odorant binding proteins or top-down regulation of the olfactory signal, etc. All in all, this is a pretty interesting paper which contributes to gain knowledge on the role of enzymes in the mechanism of olfaction.

Authors answer: We agree with the reviewer about the other aspects involved in the olfactory response which is complex. But to be as clear as possible for the readers, we chose to associate the word “response” in the text, with EOG, all along the manuscript. We changed the sentence about the olfactory response in the conclusion. 

Reviewer #3: In their manuscript "The enzyme UGT2A1, which metabolizes odors: immunolocalization and impact of the modulation of its activity on the olfactory response". Fabrice Neiers and his co-authors provide an anatomical and functional characterization of the rat UGT2A1 enzyme in the olfactory epithelium (OE). For this purpose, the authors describe a study of the localization of UGT2A1 in the OE using IHC and electron microscopy, as well as an original inhibition approach based on the use of betaglucuronidase. Overall, the manuscript is clearly written, but some of the experiments seem to be preliminary, and the manuscript would be improved by a revision. The immunolocalization of UGT2A1 in rat OE is of great interest and the potential localization of UGT2A1 in the plasma membrane of olfactory cilia is very original and constitutes a potential breakthrough in the understanding of OE expression patterns and putative function. Unfortunately, the results obtained in the present study are not convincing enough to confirm such crucial finding. 

IHC experiments are highly reminiscent of the picture obtained by Lazard et al., 1991 where a similar labelling was described as “mainly in in subepithelial Bowman’s gland” and “in a superficial epithelial layer, probably corresponding to the apical cytoplasm regions of the secretory supporting cells”. The authors should therefore provide more detailed images with higher magnification of the apical region of the OE. 

Authors answer: To our knowledge, the pioneer work from Lazard on UGT2A1 localization was obtained in bovine olfactory epithelium, the data in rat were not shown. Here, the IHC experiments are original in rat and present a much higher quality than Lazard’s work. In agreement with the reviewer, we changed the figure 1 by adding higher magnification of the apical region.

More importantly, double IHCs with neuron-specific Ab are required to conclude on putative neuronal expression. 

Authors answer: Using IHC, we did not conclude on the presence of UGT2A1 in the neurons. This localization was confirmed and observed with EM. To avoid confusion, we deleted the mention OSN in the figure 1.

The same applies to the immunogold electron microscopy study: the authors claim the use of "stringent conditions" that remains to be explained, and that the number of images is too low, especially with regard to ciliary labelling, which must be combined with a control image in a much larger microscopic study (and please indicate the number of samples treated).Moreover the intensity of the IHC labeling is not comparable with the EM regarding the relative intensities (despite stringent conditions). Thus, while this part of the study is promising, the data presented in this section require further testing to confirm the authors' conclusions.

Authors answer: 

Following the remark of the reviewer about the term “stringent conditions”, we have understood that the use of this term was confusing and not used in a proper manner, we deleted this mention. The EM technic is intrinsically stringent allowing a specific staining. To highlight the reviewer on the EM conditions optimization: two rats heads have been embedded and to obtain the best conditions, 9 condition combinations were used for (1) fixation (different chemical fixation: ratio of PFA vs Glutaraldehyde or high pressure freezing fixation), (2) resin inclusion (different resin : LR white or HM 20) and blocking condition (BSA fraction V, fish skin gelatin, acetylated BSA, powdered milk). On the 25 grids selected, the optimal dilution of the primary antibody (UGT2A1) was also optimized according to the blocking buffer. The conditions written in the materials and methods are the best conditions obtained. 

For EM in this paper, since the presence of UGT2A1 in olfactory cilia was already suggested by proteomic studies (Mayer et al. 2008, Kuhlmann et al. 2014), we wanted to precisely localize UGT2A1. We now propose additional picture showing the immunogoldstaining of UGT2A1 in the cilia. The difference in staining intensity between IHC and EM is clearly due to the EM technic itself as explain above. Moreover, in IHC, the chemistry of the staining (DAB Substrate-Chromogen) amplifies the signal. 

The authors should also consider the fact that UGT2A2 could be detected by their analysis, as the Ab is unlikely to detect a specific isoform.

Authors answer: We can not rule out that the Ab used in the study cross-react with UGT2A2 which is a splicing variant of UGT2A1. To our knowledge UGT2A2 expression and activity is much weaker in human nasal tissues compared to UGT2A1 {Court, 2012 #38}. We have added this information and the existence of a potential cross-reactivity in the corrected manuscript.

The electrophysiological approach suffers from the same caveat: the original idea of using betaglucuronidase coupled with EOG is brilliant, but this experiment requires much more analysis to conclude on the role of UGT2A1 in olfactory modulation. First traces in figure 3 are supposed to describe “typical EOG responses” but are not in line with the analysis depicted in the following graph (the number of tested animals is also not mentioned). The authors should consider traces averaging for an immediate and clear comparison of the different conditions. Moreover, in such a new a broader study should be conducted to validate the proposed approach: more odorants (guaiacol, phenylethanol, ethyl hexanol, or other aromatics…) and, more importantly, a large concentration range of betaglucuronidase to ensure its dose dependence effect on EOG responses.

Authors answer: We are grateful to reviewer 3 for his comments on our approach mixing electrophysiology and pharmacology (Fig3). This constitutes the first attempt to demonstrate that blocking the metabolization of eugenol on the olfactory epithelium leads to enhanced electrophysiological responses. More conditions will be tested in the future including what is advised by reviewer 3: additional odorants differing in their metabolic pathways and additional conditions for the betaglucuronidase administration. EOG traces showed in the figure 3A were selected because these signals were comparable in response to both odorants under control conditions (before B-Glu or vehicle treatment). A misunderstanding may have been caused by the fig 3B y-axis legend (second line). The slash symbol in the expression “before/after treatment” was not used in a proper mathematical meaning. We chose to replace it by “fold change after/before treatment” which clearly reflects the calculation that was made and we clarified the material and methods statement on this.

Regarding the data presentation, the graph from Fig 3B was done on EOG net responses i.e. response amplitudes from which blank response amplitudes were subtracted. These responses to the odorant-free solvent (distilled water in the present experiment) can be variable from one olfactory mucosa to another. This characteristic prevents a data presentation as raw EOG trace averaging. We presented our EOG data as “fold change compared to control” following a procedure published in multiple peer reviewed articles that used similar pharmacological approaches (Negroni et al., 2012; Loch et al., 2013; Savigner et al., 2009). This data analysis preserves statistical power by comparing paired samples in a before vs after treatment statistical plan. Number of tested animals has been added to the caption of figure 3.

Finally, in the discussion, the authors speculate on the potential role of UGT2A1 in modulating the olfactory response with, as I understand it, the enzyme facing the outer neuronal membrane that could buffers the amount of odorants for the OR to respond While interesting, this hypothesis is highly speculative because many of the cofactors necessary for the enzyme to function are absent from the OM. Additional experiments, such as in vitro analysis, could respond to this potential new enzymatic topology.

Authors answer: Actually, we did not mention that the enzyme or even its active site faces the outer neuronal membrane, therefore we did not propose the corresponding hypothesis in the text. However, since this point has been raised by the reviewer, to avoid any confusion, we have written differently the corresponding text. We agree with the reviewer about the question of the topology which will be of further interest for our team.

---

## [Decision Letter · Decision Letter 1]

10 Mar 2021

The odorant metabolizing enzyme UGT2A1: Immunolocalization and impact of the modulation of its activity on the olfactory response.

PONE-D-20-32295R1

Dear Dr. Heydel,

We’re pleased to inform you that your manuscript has been judged scientifically suitable for publication and will be formally accepted for publication once it meets all outstanding technical requirements.

Kind regards,

Peng He, Ph.D

Academic Editor

PLOS ONE

Additional Editor Comments (optional):

Reviewers' comments:

Reviewer's Responses to Questions

**Comments to the Author**

1. If the authors have adequately addressed your comments raised in a previous round of review and you feel that this manuscript is now acceptable for publication, you may indicate that here to bypass the “Comments to the Author” section, enter your conflict of interest statement in the “Confidential to Editor” section, and submit your "Accept" recommendation.

Reviewer #1: All comments have been addressed

Reviewer #2: All comments have been addressed

2. Is the manuscript technically sound, and do the data support the conclusions?

Reviewer #1: Yes

Reviewer #2: Yes

3. Has the statistical analysis been performed appropriately and rigorously? 

Reviewer #1: Yes

Reviewer #2: Yes

4. Have the authors made all data underlying the findings in their manuscript fully available?

Reviewer #1: Yes

Reviewer #2: Yes

5. Is the manuscript presented in an intelligible fashion and written in standard English?

Reviewer #1: Yes

Reviewer #2: Yes

6. Review Comments to the Author

Reviewer #1: The author has answered all the questions, and the article has been greatly improved. I suggest it be published.

Reviewer #2: The manuscript by Neiers et al is now acceptable for publication because my minors concerns have been addressed

7. PLOS authors have the option to publish the peer review history of their article (what does this mean?). If published, this will include your full peer review and any attached files.

Reviewer #1: No

Reviewer #2: No

---

## [Editor Report · Acceptance letter]

16 Mar 2021

PONE-D-20-32295R1 

The odorant metabolizing enzyme UGT2A1: Immunolocalization and impact of the modulation of its activity on the olfactory response. 

Dear Dr. Heydel:

I'm pleased to inform you that your manuscript has been deemed suitable for publication in PLOS ONE. Congratulations! Your manuscript is now with our production department. 

Kind regards, 

on behalf of

Dr. Peng He 

Academic Editor

PLOS ONE